# Gestational Interrelationships among Gut–Metabolism–Transcriptome in Regulating Early Embryo Implantation and Placental Development in Mice

**DOI:** 10.3390/microorganisms12091902

**Published:** 2024-09-17

**Authors:** Shuai Lin, Yuqi Liang, Jingqi Geng, Yunfei Yan, Ruipei Ding, Maozhang He

**Affiliations:** School of Basic Medical Sciences, Anhui Medical University, Hefei 230032, China; 18907757553@163.com (Y.L.); gengjq1013@163.com (J.G.); 18236850769@163.com (Y.Y.); 2113100010@stu.ahmu.edu.cn (R.D.)

**Keywords:** gestational day 8, gut microbiota, serum metabolomics, uterine transcriptome, asparagine synthetase

## Abstract

Decidualization of the uterine endometrium is a critical process for embryo implantation in mammals, primarily occurring on gestational day 8 in pregnant mice. However, the interplay between the maternal gut microbiome, metabolism, and the uterus at this specific time point remains poorly understood. This study employed a multi-omics approach to investigate the metabolic, gut microbiome, and transcriptomic changes associated with early pregnancy (gestational day 8 (E8)) in mice. Serum metabolomics revealed a distinct metabolic profile at E8 compared to controls, with the differential metabolites primarily enriched in amino acid metabolism pathways. The gut microbial composition showed that E8 mice exhibited higher alpha-diversity and a significant shift in beta-diversity. Specifically, the E8 group displayed a decrease in pathogenic Proteobacteria and an increase in beneficial Bacteroidetes and S24-7 taxa. Transcriptomics identified myriads of distinct genes between the E8 and control mice. The differentially expressed genes were enriched in pathways involved in alanine, aspartate, and glutamate metabolism, PI3K-Akt signaling, and the PPAR signaling pathway. Integrative analysis of the multi-omics data uncovered potential mechanistic relationships among the differential metabolites, gut microbiota, and uterine gene expression changes. Notably, the gene *Asns* showed strong correlations with specific gut S24-7 and metabolite L-Aspartatic acid, suggesting its potential role in mediating the crosstalk between the maternal environment and embryo development during early pregnancy. These findings provide valuable insights into the complex interplay between the maternal metabolome, the gut microbiome, and the uterine transcriptome in the context of early pregnancy, which may contribute to our understanding of the underlying mechanisms of embryo implantation and development.

## 1. Introduction

Mammalian pregnancy is a critical biological process that involves intricate interactions between the maternal and fetal units. Two key events during early pregnancy are the embryo implantation and placental development, which are crucial for fetal health and survival [1,2]. In particular, gestation day 8 (GD8) in mice is considered a critical period for these processes, involving complex molecular and cellular events, such as immune cell migration, cytokine production, and placental tissue formation. Embryo implantation, wherein the blastocyst attaches to the uterine wall, initiates the formation of the placenta—an organ vital for nutrient and gas exchange between the mother and fetus [3,4]. This event is tightly regulated by a delicate balance between maternal and embryonic signaling pathways, encompassing various hormones, growth factors, and cytokines [5,6]. Disruption of this finely tuned process can lead to implantation failure or abnormal placentation, both of which can have severe consequences for fetal development and overall pregnancy outcomes. Recent studies have highlighted the role of the microbiota and its associated metabolites in modulating these critical processes. Evidence suggests that changes in the microbiota’s composition during early pregnancy can influence maternal immune responses and metabolic pathways, potentially impacting embryo implantation and placental development. However, the precise mechanisms by which the microbiota and its metabolites affect these early pregnancy events remain poorly understood.

Strikingly, the role of the gut microbiome in maternal health and fetal development has gained increasing attention in recent years. The gut microbiota not only impacts the host’s immune system, but may also influence the interactions between the mother and the fetus through various mechanisms [7,8,9]. For instance, alterations in the gut microbiome have been associated with preterm birth, low birth weight, and other pregnancy-related complications [10,11,12]. The gut microbiome can regulate host metabolism, immune responses, and endocrine function, all of which are critical for a healthy pregnancy [13,14]. During pregnancy, the maternal gut microbiome undergoes dynamic changes, which may have downstream effects on embryo implantation and placental development [15]. For example, the gut microbiome can modulate the production of metabolites, such as short-chain fatty acids, that can influence the uterine receptivity and placental function [12,16]. Furthermore, the maternal gut microbiome–host interactions may also involve changes in the gene expression patterns in the maternal organs, such as the uterus and placenta, which are crucial for supporting fetal growth and development [17,18]. Understanding the interplay between the maternal gut microbiome, the metabolome, and the transcriptome during critical stages of pregnancy, such as embryo implantation and placental development, may provide important insights into the mechanisms underlying maternal–fetal crosstalk and inform the development of new pregnancy intervention strategies.

Metabolomics, the study of the comprehensive profile of and dynamic changes in all small molecules (e.g., amino acids, fatty acids, carbohydrates, etc.) within a biological system, has emerged as a powerful tool to investigate the interactions between the gut microbiome and the host metabolism [19,20]. Previous research has indicated that early pregnancy induces significant changes in the plasma metabolome [21]. Abnormal metabolite profiles during the early stages of pregnancy have been identified as potential predictive biomarkers for a variety of gestational diseases [22,23,24]. In addition, changes in the maternal metabolome may reflect alterations in the gut microbiome’s composition and function, which, in turn, can impact the health of the mother and the developing fetus [25,26]. By integrating gut microbiome, metabolomic, and transcriptomic analyses, researchers can gain a more comprehensive understanding of the complex interplay between the maternal gut–metabolism–transcriptome axis and its influence on critical stages of pregnancy, such as embryo implantation and placental development.

Given the considerations outlined above, the present study aims to investigate the potential associations between the maternal gut microbiome, metabolome, and transcriptome during the critical gestational D8 in mice. Our goal is to elucidate the interrelationships between these factors and explore their underlying biological mechanisms, with the ultimate aim of providing a theoretical foundation for future clinical interventions.

## 2. Materials and Methods

### 2.1. Ethics and Consent

All the animals were raised in compliance with the care and use guidelines of experimental animals according to the Animal Welfare legislation of China. This study was approved by the Ethics Committee of Anhui Medical University.

### 2.2. Animals and Samples Collection

A total of 12 female C57BL/6J mice (comprising six that were identified pregnant for eight days, and six that were non-pregnant, serving as the control group) were included in this study. All animals were fed identical autoclaved diets and water ad libitum and were maintained under specific pathogen-free conditions with a 12 h light/12 h dark cycle, temperature of 24 ± 2 °C, and humidity of 50% ± 10%. After euthanizing the mice with isoflurane, blood was collected by retro-orbital bleed to serum-separation collection tubes. Colon luminal contents were carefully collected from mice and transferred to sterile tubes. Uterine tissue from the implantation sites of the pregnant mice was excised, while corresponding uterine tissue was also collected from the control group. Following collection, blood samples were centrifuged (5000 rpm, 4 °C for 10 min) and the supernatant was transferred into a new tube. At last, all of the samples were rapidly frozen in liquid nitrogen and stored at −80 °C in a freezer for storage and further processing.

### 2.3. RNA-Seq and Differential Gene Expression Analysis

RNA sequencing (RNA-seq) and subsequent differential gene expression analysis were conducted following established protocols as detailed in previous research [27]. Briefly, total RNA was extracted from uterus tissues using a Trizol kit (Invitrogen, Waltham, MA, USA) following the manufacturer’s protocol. cDNA libraries were then constructed from the extracted RNA using the NEBNext^®^ Ultra™ Directional RNA Library Prep Kit for Illumina^®^ (NEB, Ipswich, MA, USA). The prepared libraries were subjected to paired-end sequencing on a HiSeq 4000 platform (Illumina, San Diego, CA, USA) by Personalbio (Personalbio, Shanghai, China), generating approximately 46.9 million 150 bp paired-end reads. Following sequencing, we employed quality trimming and filtering to remove low-quality reads and adaptors using Cutadapt (version 2.8). Differential expression gene (DEG) analysis was conducted using DESeq2 package (version 1.44.0) [28], identifying significantly expressed genes based on a threshold of adjusted *p* values less than 0.05 and absolute log2 (fold change) >1. The functions and pathways of DEGs were annotated by the GO and KEGG databases via clusterProfile R package [29].

### 2.4. Microbial DNA Extraction and 16S rRNA Gene Sequencing

Following established methodologies [30], microbial DNA was extracted, and the 16S rRNA gene was subsequently sequenced. This sequencing was performed on an Illumina MiSeq platform, adhering to the manufacturer’s guidelines (PANO-MIX Bio-Pharm Technology Co., Ltd., Soochow, China). The resulting data underwent splicing and filtering with Trimmomatic (version 0.35) [31], producing a collection of high-quality sequences for further analysis. The reads were clustered into operational taxonomic units (OTUs) at a similarity threshold of 97.0% using USEARCH (version 10.0) [32]. For species classification, feature sequences were annotated with a Naive Bayes classifier, referencing the database provided by Silva (http://www.arb-silva.de, accessed on 16 December 2019) [33]. Finally, alpha diversity indices, including Chao1 and Shannon, as well as beta diversity calculated with the Bray–Curtis dissimilarity matrix, were analyzed using Quantitative Insights into Microbial Ecology (QIIME) (version 1.9.1). Additionally, principal component analysis (PCA) based on the Bray–Curtis matrix was performed using the vegan package.

### 2.5. Nontargeted Serum Metabolite Profiling and Analysis

The metabolite profiles of serum were detected by an LC-MS/MS system consisting of an ultra-high-performance liquid chromatography (UHPLC) system (Vanquish, Thermo Fisher Scientific, Waltham, MA, USA) coupled to a Q Exactive (QE) HFX mass spectrometer (Orbitrap MS, Thermo Fisher Scientific, Waltham, MA, USA). The UHPLC system was equipped with a UPLC BEH Amide column (2.1 mm × 100 mm, 1.7 μm) and was used for the separation of biochemicals. The mobile phase consisted of 25 mmol/L ammonium acetate (Sigma-Aldrich, Darmstadt, Germany) and 25 mmol/L ammonium hydroxide (Fisher Chemical, Waltham, MA, USA) in water (pH = 9.75) (solvent A) and acetonitrile (solvent B). The autosampler temperature was maintained at 4 °C, and the injection volume was 3 μL. The QE HFX mass spectrometer was operated in information-dependent acquisition (IDA) mode to obtain MS/MS spectra, under the control of the Xcalibur acquisition software (version 1.2, Thermo Fisher Scientific, Waltham, MA, USA). In this mode, the acquisition software continuously evaluates the full-scan MS spectrum. The electrospray ion (ESI) conditions were as follows: sheath gas flow rate of 30 Arb, auxiliary gas flow rate of 25 Arb, capillary temperature of 350 °C, full MS resolution of 60,000, MS/MS resolution of 7500, collision energy of 10/30/60 in negative ion mode (NCE), and spray voltage of 3.6 kV (positive) or −3.2 kV (negative). Next, the original LC-MS/MS data were converted to the mzXML format using ProteoWizard (version 3.0) and processed with an in-house program, which was developed using R and based on XCMS, for peak detection, extraction, alignment, and integration. Subsequently, for further multivariate analysis, the raw data were pretreated. Pretreatment included de-noising based on the relative standard deviation (RSD), filling the missing data via half of the minimum value, and normalization by the internal standard normalization method. The final dataset contained the information of the peak number, sample name, and normalized peak area and was imported to SIMCA16.0.2 software package (Sartorius Stedim Data Analytics AB, Umea, Västerbotten, Sweden). Firstly, principal component analysis (PCA), an unsupervised method, was performed to reduce the dimensionality of the data. The 95% confidence interval (95% CI) in the PCA score plot was used as the threshold to identify potential outliers in the dataset. Secondly, in order to visualize the group separation and find significantly changed metabolites, we conducted supervised orthogonal projections to latent structure discriminate analysis (OPLS-DA) and acquired the value of variable importance in projection (VIP). A 7-fold cross-validation test was performed to evaluate the goodness-of-fit of the OPLS-DA model using the values of R^2^Y and Q^2^. R^2^Y indicates how well the variation of a variable is explained, and Q^2^ indicates how well a variable can be predicted. A 200-time permutation test was then conducted to assess the robustness of the model. The precursor molecule passed the combined criteria: (1) absolute log2 fold change (|log2FC|) greater than 0.26, (2) VIP > 1, and (3) coarse *p* value < 0.05 (nonparametric Wilcoxon *t*-test) were considered significantly changed between mice in E8 and sham groups. For clustering heatmaps, the data were normalized using z-scores of the intensity areas of differential metabolites and were plotted by Pheatmap package in R language (version 4.4.0). Furthermore, after identifying the differential metabolite features, the fragment information obtained from the MS/MS analysis was matched with annotations from HMDB, Metlin, LipidMaps, and an in-house standards database (PPM < 10) to ensure accurate metabolite identification. Finally, commercial databases, including the Kyoto Encyclopedia of Genes and Genomes (KEGG) (http://www.genome.jp/kegg/; accessed on 15 June 2021) and MetaboAnalyst (http://www.metaboanalyst.ca/; accessed on 15 June 2021), were employed for pathway enrichment analysis. Metabolic pathways were considered statistically significant when the *p*-value was less than 0.05.

### 2.6. Random Forest Model for Selection Pregnant Markers

To identify gut microbiota and metabolites that can distinguish early pregnancy from sham controls, a random forest model was constructed with modified settings (mtry = 39, ntree = 600). The optimal number of species was determined using 10-fold cross-validation with the rfcv function from the random forest package. The most highly discriminating species were identified based on importance values characterized by the “MeanDecreaseAccuracy” parameter.

### 2.7. Statistical Analysis

The present study adhered to rigorous statistical methods to ensure the validity and reliability of the results. All data were analyzed by R language software (Version 4.0.3) and expressed as mean ± Sd. The Shapiro–Wilk test was utilized to determine the data-distribution type. For the two groups, after determining that the data were not normally distributed and their homoscedasticity, two-sided but unpaired Mann–Whitney tests were used to calculated the statistical significance. To control for the false discovery rate (FDR), multiple testing was corrected using the Benjamini–Hochberg method, considering FDR-corrected *p* value of less than 0.05 as statistically significant. Log_10_-transformed metabolite dataset was utilized to construct modules via soft-threshold Pearson correlation analysis, in combination with a topological overlap distance metric and average hierarchical clustering, using the weighted correlation network analysis (WGCNA) in the R package. Further, the relationship between differential omics was calculated using the Spearman rank correlation.

## 3. Results

### 3.1. Early Pregnant Mice Show a Distinct Serum Metabolic Profile Compared with Mice in Sham Group

The analysis conducted in this study identified significant metabolic changes associated with the stimulant introduced by early pregnancy. An untargeted global metabolic profiling was conducted on serum samples from six mice at eight days post-pregnancy (E8) and six control (sham) mice. Typical extracted ion chromatograms (base peak chromatogram) from two ESI modes are presented in Appendix A. In the principal component analysis (PCA) score plots of both ESI^+^ and ESI^−^, the quality control (QC) samples were observed to cluster tightly together, indicating the reliability of the present study. Additionally, the coefficients of the variation in peak distribution in the QC samples confirmed that the analysis was stable and repeatable (Appendix A). The metabolomic analysis revealed a distinct serum metabolic profile in the E8 mice compared to the sham group, as demonstrated by the PCA score plots in positive and negative ion modes (Figure 1A,B). Of the 351 annotated metabolites, a total of 72 were found to be significantly altered between the two groups, as revealed by Welch’s two-sample *t*-test. Among the altered metabolites, 31 (43.06%) were enriched in the E8 mice, while 41 (56.94%) were downregulated (Figure 1C, Appendix A). To further delineate potential correlations with early pregnancy, individual serum metabolites were subjected to in-depth analyses. A random forest (RF) analysis was performed, which ranked the altered 72 serum metabolites by their contribution to the group separation. The top 19 differentiating metabolites with a mean decrease in their Gini index greater than 0.1 are shown in Appendix A. In addition, KEGG enrichment analysis based on the differential metabolites was conducted to reveal altered biological processes and demonstrated the functional differences of the metabolites. The differential metabolites induced by pregnancy were mainly enriched in “Valine, leucine and isoleucine biosynthesis”, “Arginine biosynthesis”, and “Alanine, aspartate and glutamate metabolism” (Figure 1D). Taken together, these findings show a sub-metabolic pathway from N-Acetyl-L-aspartic acid to Ureidosuccinic acid within the alanine, aspartate, and glutamate biosynthesis pathways and showed that aspartate N-acetyltransferase, aspartoacylase, and aspartate carbamoyltransferase are required (Figure 1E). These findings suggest that significant metabolic alterations occur in E8 mice, potentially reflecting changes in key biological processes and metabolic pathways associated with early pregnancy. Future research should investigate potential interventions that could modulate these metabolic changes, which may provide insights into managing pregnancy-related complications and enhancing maternal health.

### 3.2. Gut Microbiota Composition Was Altered in E8 Mice

Microbial analysis revealed significant differences in the gut microbial composition between the pregnant E8 mice and sham C57BL/6J mice. Using 16S rRNA gene sequencing, we compared cecum contents from both groups. The E8 mice demonstrated a relatively higher α-diversity, and exhibited a marked shift in microbial composition when compared to sham mice, as evidenced by a significant difference in β-diversity (*p* = 0.002) (Figure 2A,B). The relative abundances of pathogenic Proteobacteria at the phylum level and *Rikenella* at the genus level were decreased in the E8 group. In addition, the relative abundances of Bacteroidetes and S24-7 were enriched in the E8 mice (Figure 2C,D). At the ASV level, we detected 22 bacteria that were significantly different between the E8 and sham groups. Nine unidentified S24-7 taxa (ASV104, ASV40, ASV173, ASV67-68, ASV217, ASV14, ASV34, and ASV168), two unidentified Rikenellaceae (ASV35, and ASV71), and *Lactobacillus vaginalis* were decreased (Figure 2E, Appendix A). Subsequently, RF analysis was performed to examine the ability to discriminate samples from E8 or sham mice based on the differential gut microbiota. A sum of 13 bacteria was shown in Appendix A, among which the ASVs that belonged to S24-7 could be prominent microbial biomarkers that were associated with the early pregnancy of mice. Furthermore, we compared the Firmicutes/Bacteroidetes ratio, a measure of the relative abundance of Firmicutes and Bacteroidetes bacteria in the gut microbiome which has been suggested to indicate overall gut health. The results exhibited a lower Firmicutes abundance in subjects of the E8 group (Figure 2F), indicating the pivotal role of embryo implantation and placental development in shaping the host gut microbial composition. These results suggest that changes in the gut microbiota may be closely correlated with the physiological changes occurring during early pregnancy. Future research could explore interventions, such as dietary modifications or probiotic supplementation, to further investigate their effects on the gut microbiota composition and overall maternal health during pregnancy. Such interventions may help to optimize gut health and mitigate potential pregnancy-related complications.

### 3.3. The Transcriptome of Gravid Uterus in E8 Mice Was Distinct Compared to Un-Pregnant Mice

The findings of this study highlight the specific mechanisms by which early pregnancy reshapes the gene expression profile from a transcriptomic perspective. In this part, we conducted RNA sequencing (RNA-seq) on uterine samples from the E8 and sham groups of mice. Firstly, the PCA result showed that the samples were clearly separated by group, which indicated the reliability and validity of our data, laying the foundation for further analysis (Figure 3A). The differentially expressed genes (DEGs) between the E8 and sham individuals were then identified with the criteria |log2 fold change| ≥ 1 and *p* value < 0.05 (Appendix A). we labeled the top 20 significantly upregulated and downregulated DEGs in the indicated groups (Figure 3B). Next, KEGG enrichment analyses were employed and, interestingly, found that the DEGs between the E8 and sham groups were significantly enriched in the “Protein processing in endoplasmic reticulum”, “PI3K-Akt signaling pathway”, and “PPAR signaling pathway” (Figure 3C, Appendix A). Interestingly, GO enrichment analysis found that most of the upregulated genes induced by embryo implantation fall into the broad categories of regulation of vasculature development and regulation of angiogenesis (Figure 3D,E, Appendix A). Taken together, the above results revealed that embryo implantation may mediated the development of the embryo through the modulation of gene expression in the uteri of the pregnant mice.

### 3.4. The Putative Mechanistic Correlations among Serum Metabolites, Gut Microbiota and Uterus Transcriptome

The multi-omics nature of this dataset provides a valuable opportunity to explore the associations among metabolites, microbes, and genes. Spearman’s correlations were conducted for the differentially abundant bacteria, metabolites, and genes, specifically focusing on the E8 and sham groups. As a result, 533 associations between species and metabolites were identified as significantly different (*p* < 0.05) (Figure 4A, Appendix A). Weighted gene co-expression network analysis (WGCNA) revealed three modules based on differential metabolites. Notably, E8-depleted unclassified Rikenellaceae were positively correlated with the blue and brown modules, while showing an inverse association with the turquoise module (Figure 4B, Appendix A). Shedding light on the core interaction among genes, metabolites, and the gut microbiota plays pivotal role in clarifying the molecular connection with early embryo implantation in the uterus. An integrated analysis of multi-omics data, including transcriptomics, metabolomics, and the colon microbiota, was employed to gain insights into whether specific genes align with distinct metabolites and microbes that contribute to the potential mechanisms of embryo development. To perform this integrative analysis, the Mantel test was utilized, incorporating differentially expressed genes (DEGs) from alanine, aspartate, and glutamate metabolism (Figure 5A), significantly altered metabolites from metabolomics (Figure 1C), and significantly different genera from gut metagenomics (Figure 2E). Distance-corrected dissimilarities of metabolites and genera were initially correlated with those of genes (Figure 5B). Overall, *Asns* exhibited strongest correlations with both microbiota and metabolites. Specifically, a chord diagram was employed to illustrate one-to-one interactions between genes and gut microbiota (Figure 5C), as well as between genes and metabolites (Figure 5D and Appendix A).

## 4. Discussion

Growing evidence suggests that dynamic changes in the maternal gut microbiome during pregnancy are crucial for fetal development [34,35]. Alterations in maternal gut microbiome diversity have been observed in early pregnancy, accompanied by concomitant changes in the maternal blood metabolome. These changes may influence early embryo implantation and normal fetal development through the modulation of host immunity, inflammatory responses, and metabolic states [7,8]. Increased abundances of certain bacterial genera, such as *Lactobacillus* and *Bifidobacterium*, in the maternal gut during pregnancy have been associated with healthy fetal development, optimal birth weight, and enhanced immune function [36,37]. Conversely, the overgrowth of pathogenic bacteria has been linked to pregnancy complications, including early pregnancy loss and preeclampsia [10,11,12,15,18]. Furthermore, changes in the maternal gut metabolome, such as altered levels of short-chain fatty acids and amino acids, may regulate the intrauterine environment by influencing the metabolism of maternal endocrine and energy, thereby impacting embryonic and fetal development [16]. However, the existing research has primarily focused on late pregnancy, while the changes in the gut microbiome, metabolome, and uterine transcriptome during early gestation, as well as the molecular interactions between these factors, remain relatively unexplored. Therefore, it is crucial to identify the key gut bacteria and circulating metabolites that influence embryo implantation during early pregnancy. Elucidating these factors could provide valuable insights into the mechanisms by which the gut microbiome, metabolism, and gene expression interact to influence early embryo establishment. In the present study, significant disturbances were observed in the serum metabolome, gut microbiota, and gene expression of the E8 mice compared to control mice. Key findings included a significant decrease in serum L-aspartic acid levels in the E8 group and a notable enrichment of the amplicon sequence variant (ASV) corresponding to the gut microbiome family S24-7. The differentially abundant metabolites and genes were concurrently enriched in the alanine, aspartate, and glutamate metabolism pathway. Specifically, the *Asns* gene within this pathway showed a significant negative correlation with L-aspartic acid levels, while exhibiting a significant positive correlation with the abundance of S24-7.

Previous research has proposed that dominant changes in serum metabolites, particularly those classified as amino acids and lysophosphatidylcholines (LPCs), are induced by early pregnancy [38]. In normal pregnancies, the levels of most serum amino acids are known to increase, which may play a pivotal role in supporting fetal health and growth [39]. In the present study, it was found that the majority of the serum metabolites that were increased or decreased in the E8 mice were classified as amino acids. Notably, L-aspartic acid and phenylacetic acid exhibited a decreasing trend in the serum of mice at eight days post-implantation. The significantly reduced levels of L-aspartic acid in the early pregnant mice represent, to the best of our knowledge, a novel finding. This reduction may reflect an active self-protective mechanism employed by the mother, as L-aspartic acid has been associated with gestational diabetes mellitus (GDM), a condition that can adversely affect both maternal and fetal health [40]. Furthermore, L-aspartic acid has been implicated in neurotoxicity in newborn mice [41], which has led to recommendations against its supplementation during pregnancy and lactation [42]. Thus, we propose that the decreased serum levels of L-aspartic acid in early pregnancy mice may serve as an innate protective mechanism, allowing gestating females to safeguard both themselves and their developing offspring. This finding underscores the importance of further investigating the implications of serum metabolite changes during early pregnancy, as well as their potential roles in maternal health and fetal development. A more comprehensive discussion on these mechanisms will not only enhance this manuscript’s clarity but also contribute to a deeper understanding of metabolic adaptations during early gestation.

The research findings of this study indicate that the depletion of the maternal gut microbiome significantly impacts the growth and development of the placenta [36], suggesting that a healthy maternal gut microbiome is crucial for the mother’s gestational period and fetal development. However, the studies investigating changes in the maternal gut microbiome during early pregnancy remain limited. This study, therefore, focused on the composition of the maternal gut microbiome in pregnant mice at gestational day 8. The most notable change observed was an increase in the abundance of the S24-7 family. Bacteria within the S24-7 (phylum Bacteroidetes) family are dominant in the mouse gut microbiota and have also been detected in the intestine of other animals. Recently, the name Muribaculaceae has been adopted to substitute S24-7 [43]. Previous research has indicated that the increased abundance of S24-7 is negatively correlated with the production of inflammatory cytokines [44], suggesting a potential role in modulating the immune response. In addition, it has been proposed that S24-7 serves as a major utilizer of mucin glycans, while many enteric pathogens depend on sugars as a nutrient source. Thus, the increased abundance of S24-7 during early pregnancy may effectively inhibit the growth of pathogenic bacteria by occupying their ecological niche. This mechanism could be crucial for preventing pathogen infection during the early stages of pregnancy [45].

In the current study, a number of differential metabolites and differentially expressed genes were found to be enriched in the same KEGG pathway: alanine, aspartate, and glutamate metabolism. Among the genes in this pathway, the *Asns* gene showed the strongest correlation with metabolomics and 16S rRNA amplicon data. *Asns* is a key enzyme in converting L-aspartic acid into asparagine [46]. *Asns* deficiency or mutation in the parents has been reported to cause a number of pediatric neurological and developmental abnormalities, such as intellectual disability, developmental delays, and asparagine synthetase deficiency (ASD) [47,48,49]. Hence, the fact that increased expression of the *Asns* gene and decreased levels of L-aspartic acid have been observed suggests the existence of a reciprocal regulation between maternal genetics and metabolism during early pregnancy. The upregulation of *Asns* may help prevent the occurrence of adverse pregnancy outcomes and ensure normal fetal development. This finding suggests that modulating the *Asns* expression or L-aspartic acid levels could be a promising strategy for managing pregnancy-related conditions. For instance, future research should confirm whether enhancing *Asns* activity or reducing the level of L-aspartic acid could positively influence pregnancy outcomes, and this could lead to the development of targeted nutritional or pharmacological interventions. Moreover, the role of the gut microbes, particularly the S24-7 genus, in regulating *Asns* expression and L-aspartic acid levels highlights the potential for microbe-based therapies. Probiotic or prebiotic interventions that promote beneficial gut bacteria could be investigated as a means to support healthy pregnancy outcomes.

## 5. Conclusions

In summary, our study has identified changes in the gut microbial composition, serum metabolic profiles, and uterine gene expression in early pregnant mice. Notably, the increased abundance of the S24-7 genus and elevated *Asns* gene expression were found to be inversely correlated with levels of L-aspartic acid. These findings suggest that S24-7, *Asns*, and L-aspartic acid could serve as potential targets for interventions using traditional Chinese medicine to address pregnancy-related diseases and fetal developmental disorders. To enhance the impact of these findings, several future research directions could be pursued. First, validating these results in human studies would be critical. This could involve analyzing the gut microbiome and serum metabolite profiles of pregnant individuals at similar gestational stages to assess the relevance of our mouse-model findings. Additionally, integrating other omics analyses, such as proteomics or relevant posttranslational modification omics, could provide deeper insights into the biological pathways involved. For instance, proteomic studies could help elucidate the functional roles of proteins associated with the identified metabolites and microbial populations. Furthermore, investigating the mechanistic pathways linking S24-7, *Asns*, and L-aspartic acid could involve in vitro studies or animal models that manipulate these factors to observe their effects on pregnancy outcomes. Exploring the influence of dietary interventions or specific herbal compounds from traditional Chinese medicine on these targets may also yield valuable insights. Overall, these directions would not only validate our findings but also expand our understanding of the interplay between the microbiome, metabolism, and gene expression during early pregnancy.

## Figures and Tables

**Figure 1 microorganisms-12-01902-f001:**
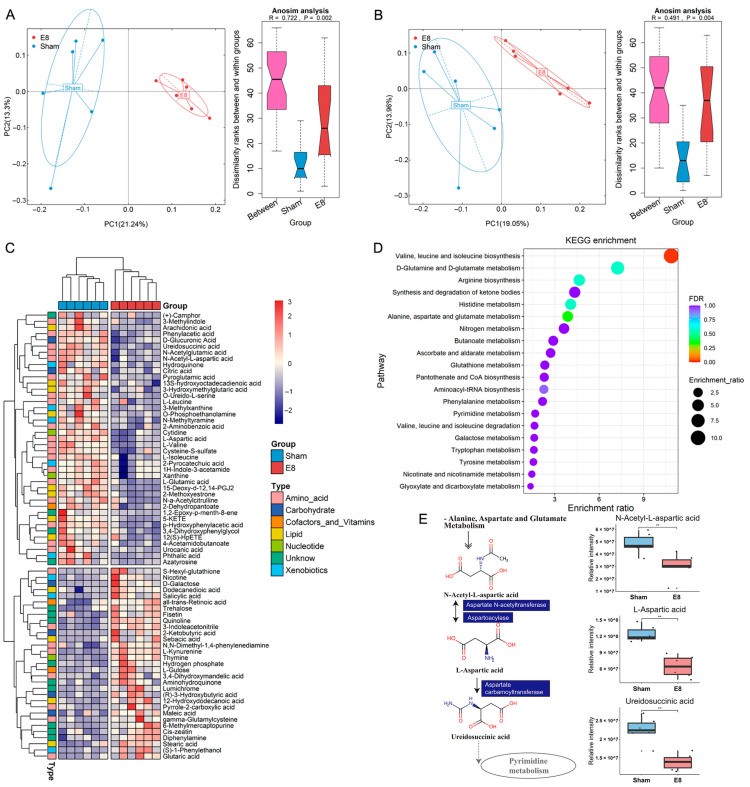
Serum metabolites were significantly changed in E8 group as compared to sham group. (**A**,**B**) Principal coordinate analysis (PCoA) based on Bray–Curtis dissimilarity calculated by metabolite features in ESI^+^ and ESI^−^ mode indicating an obvious shift in mice of E8 group from that in sham group. (**C**) Heatmap clustering of distinct serum metabolites from the comparison between E8 and sham groups. (**D**) KEGG pathway analysis was performed based on differentially expressed metabolites between E8 and sham groups. The color of bubbles represents the value of adjusted *p* value, and the size of bubbles represents the number of counts (sorted by enrichment ratio). (**E**) Schematic diagram of 3 E8-depleted metabolites participating in the alanine, aspartate, and glutamate metabolism KEGG pathways. (Asterisk indicates statistical significance, ‘**’ represents *p* < 0.01).

**Figure 2 microorganisms-12-01902-f002:**
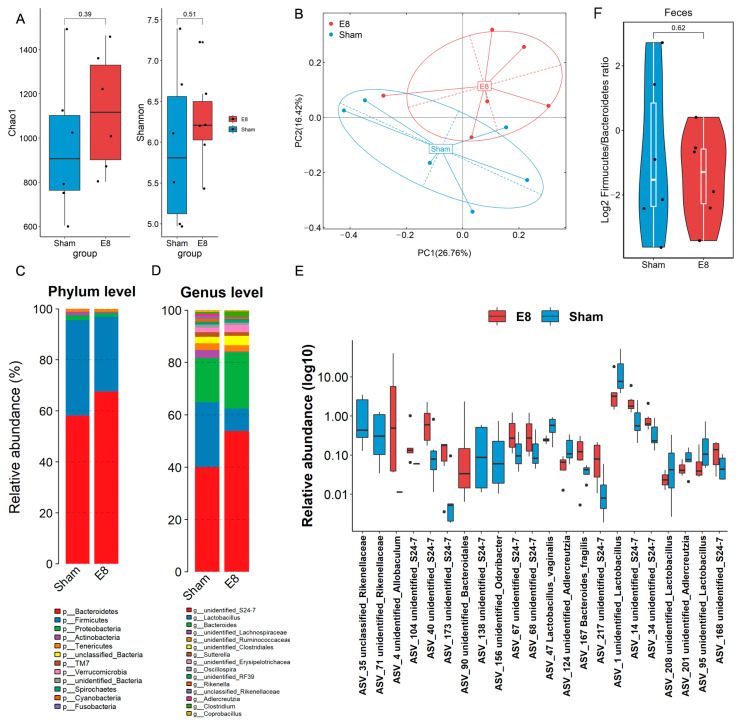
Compositional analysis of gut microbiota in mice between sham and E8 groups. (**A**) Faecal alpha diversity analysis estimated by Chao1 and Shannon indexes between mice in sham and E8 groups. (**B**) Principal coordination analysis (PCoA) and analysis of similarities (anosim) were calculated based on Bray–Curtis matrix. (**C**,**D**) The taxonomic composition distribution between two groups on the phylum and genus levels of gut microbiota. (**E**) Log10-transformed relative abundance of significantly different ASVs between sham and E8. (**F**) The ratio of Firmicutes compared to Bacteroidota in E8 mice compared with sham mice.

**Figure 3 microorganisms-12-01902-f003:**
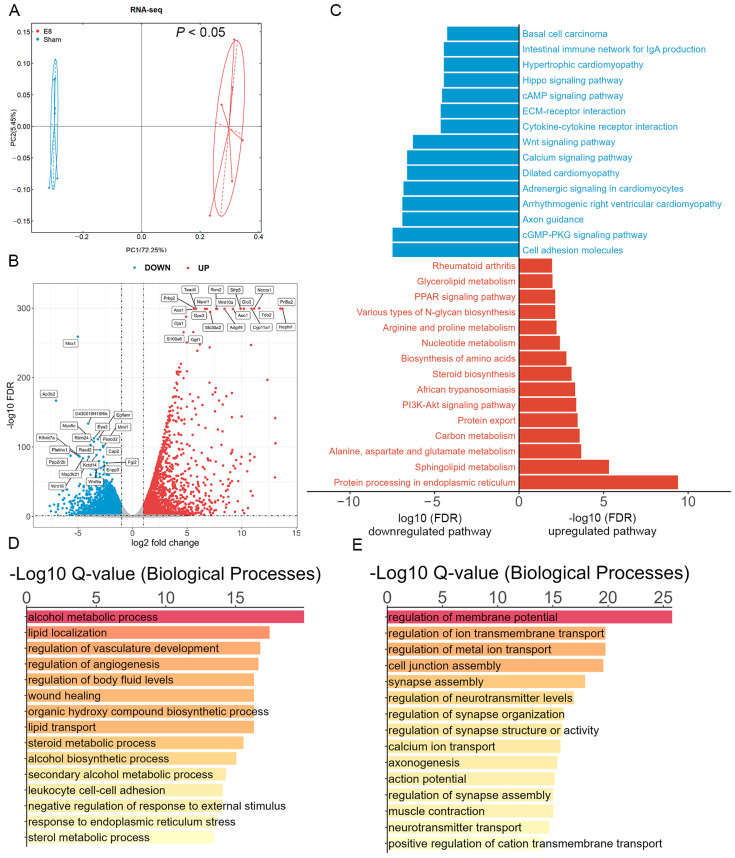
Analysis of differentially expressed genes in E8 and sham groups. (**A**) PCA score plot indicating an obvious separation of the transcriptomic profile between E8 and sham groups. (**B**) Volcano plot showing differential genes (top 20 upregulated and top 20 downregulated genes in E8 and sham mice). Criteria for significant differences (VIP > 1, adjusted *p* < 0.05 and fold change ≥ 2). (**C**) KEGG enrichment analysis of the DEGs in the E8 versus sham comparison. The red column represents the top 15 KEGG pathways in E8 and blue column represents the top 15 downregulated KEGG pathways. (**D**,**E**) The bar plot for the top 15 significant enrichment annotations for biological process (BP) of all upregulated genes (**D**) and downregulated genes (**E**).

**Figure 4 microorganisms-12-01902-f004:**
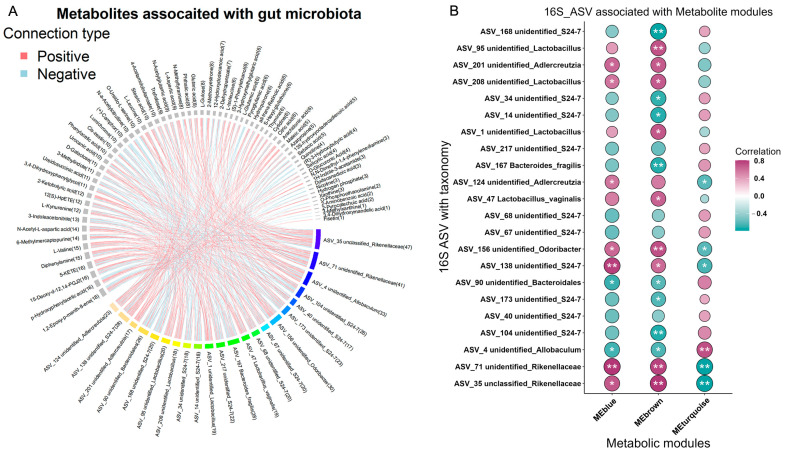
Associations between differentially expressed metabolites and gut microbiota. (**A**) Chord diagram displaying the significant associations between differentially expressed metabolites and ASVs. The associated metabolites are colored gray, and the associated ASVs are colored successively. Each line indicates a significant correlation between a bacterium and a metabolite, with the red color corresponding to a positive association (*p* value < 0.05) and the blue color representing a negative association (*p* value < 0.05). (**B**) Correlations between metabolite modules and ASVs. The absolute correlation coefficient (|r|) corresponds to the size of the circle, and the *p* value is indicated by asterisk (“*”, *p* < 0.05; “**”, *p* < 0.01).

**Figure 5 microorganisms-12-01902-f005:**
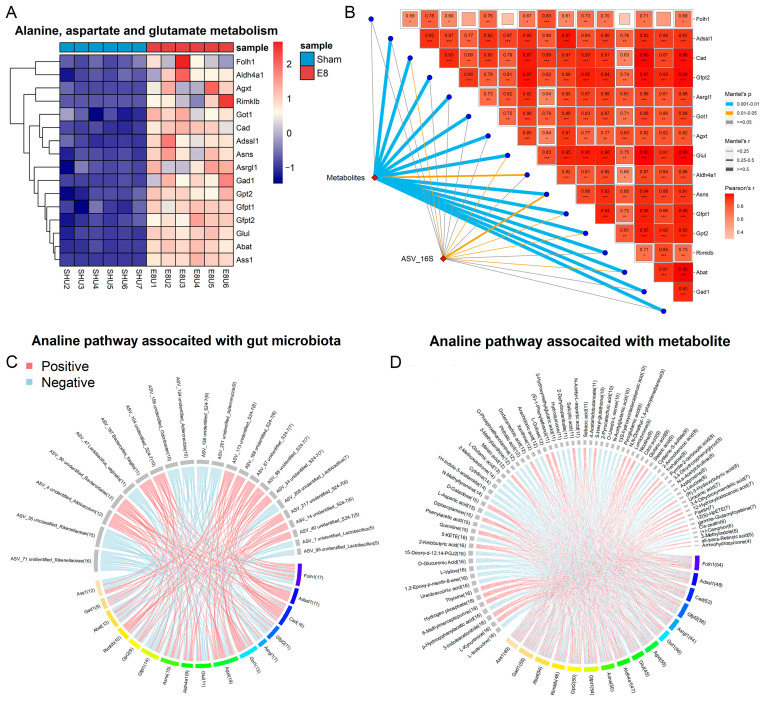
Variations in serum metabolites and gut microbiota have extensive relationship with pregnancy-related KEGG pathways. (**A**) Heatmap shows differentially expressed genes in the alanine, aspartate, and glutamate metabolism pathways. (**B**) 16S rRNA gene sequencing based on ASVs composition and metabolites composition was related to each of 16 significantly differential genes in the alanine, aspartate, and glutamate metabolism pathways using Mantel-test analysis. Edge width corresponds to the Mantel’s r statistic for the corresponding distance correlations, and edge color denotes the statistical significance. Pairwise comparisons of genes are shown, with a color gradient denoting Spearman’s correlation coefficient (Asterisks in the cells indicate statistical significance of the pairwise correlation, ‘*’ represents *p* < 0.05, ‘**’ represents *p* < 0.01, ‘***’ represents *p* < 0.001). (**C**) Chord diagram displays the significant associations between each ASV and gene in the alanine, aspartate, and glutamate metabolism pathway. The associated genes are colored gray, and the associated ASVs are colored successively. Each line indicates a significant correlation between a bacterium and a metabolite, with the red color corresponding to a positive association (*p* value < 0.05) and the blue color representing a negative association (*p* value < 0.05). (**D**) Chord diagram displays the significant associations between each metabolite and gene in the alanine, aspartate and glutamate metabolism pathway. The associated genes are colored gray, and the associated metabolites are colored successive. Each line indicates a significant correlation between a bacterium and a metabolite, with the red color corresponding to a positive association (*p* value < 0.05) and the blue color representing a negative association (*p* value < 0.05).

## Data Availability

The raw data supporting the conclusions of this article will be made available by the authors on request. Annotated metabolomic data of serum is available from Appendix A.

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
