# Peer review of "Gestational Interrelationships among Gut–Metabolism–Transcriptome in Regulating Early Embryo Implantation and Placental Development in Mice"

_microorganisms, 2024, doi:10.3390/microorganisms12091902_

Round 1

Reviewer 1 Report

Comments and Suggestions for Authors

The manuscript titled "Gestational interrelationships among gut-metabolism-transcriptome in regulating early embryo implantation and placental development in mice" presents valuable insights into the complex interplay between the maternal metabolome, gut microbiome, and uterine transcriptome in the context of early pregnancy. The study effectively integrates metabolomics, gut microbiota analysis, and transcriptomics;  correlation between Asns gene expression, S24-7 microbiota, and L-Aspartic acid levels presents a new insight that could significantly impact the field.

Several ways of improvement would help make the manuscript more impactful and accessible, potentially broadening its appeal to a wider audience.

The results are comprehensive but could be more clearly structured to highlight the most significant findings. For example, summarizing the key results in a brief paragraph at the beginning of each subsection could help readers grasp the main points before delving into the detailed data.

The discussion should delve deeper into the implications of the findings, particularly how they could influence future research or clinical practices. For instance, the significance of the Asns gene and its potential as a target for clinical interventions could be explored more thoroughly.

While the conclusion summarizes the findings well, it could be enhanced by suggesting specific future research directions that stem from this study. For example, how might these findings be validated in human studies, or what other omics analyses could be integrated?

Overall, I would like to congratulate authors with a decent work and recommend manuscript for publication after minor editions (courtesy of editor).

Author Response

# Reviewer 1

The manuscript titled "Gestational interrelationships among gut-metabolism-transcriptome in regulating early embryo implantation and placental development in mice" presents valuable insights into the complex interplay between the maternal metabolome, gut microbiome, and uterine transcriptome in the context of early pregnancy. The study effectively integrates metabolomics, gut microbiota analysis, and transcriptomics; correlation between Asns gene expression, S24-7 microbiota, and L-Aspartic acid levels presents a new insight that could significantly impact the field.

Several ways of improvement would help make the manuscript more impactful and accessible, potentially broadening its appeal to a wider audience.

Response: We really appreciate for your thoroughly examination and valuable suggestions to this work, we will check our manuscript throughout carefully and revised it point by point and use blue font to indicate the modified part.

The results are comprehensive but could be more clearly structured to highlight the most significant findings. For example, summarizing the key results in a brief paragraph at the beginning of each subsection could help readers grasp the main points before delving into the detailed data.

Response: Thank you for this advice. The part of “results” has been re-structured in order to make it more readable.

The discussion should delve deeper into the implications of the findings, particularly how they could influence future research or clinical practices. For instance, the significance of the Asns gene and its potential as a target for clinical interventions could be explored more thoroughly.

Response: Thank you for the reviewer's suggestions. We have added a discussion about future research in the discussion section.

While the conclusion summarizes the findings well, it could be enhanced by suggesting specific future research directions that stem from this study. For example, how might these findings be validated in human studies, or what other omics analyses could be integrated?

Response: Thanks for this suggestion. We have revised this part in the revised manuscript.

Overall, I would like to congratulate authors with a decent work and recommend manuscript for publication after minor editions (courtesy of editor).

Reviewer 2 Report

Comments and Suggestions for Authors

Dear authors,

Despite the interesting topic provided in the title, unfortunately there are way too serious issues with the manuscript:

1. In the intoductions there is a lack of a problem presented so that the current research is justified. In order to prove your point you should provide enough robus evidence that the microbiota and associated changes of metabolites lead to any changes in early pregnancy, and that furher studies in that topic are needed.

2. As a practicing clinician my biggest problem is that there is absolutely no discussion or evidence on how would this research translate to clinical practice. 

3. The results section does not explain how the findings may be correlated to changes in early pregnancy and what if any interventions would be able to affect them.

4. You have not followed MDPI's guidelines on citing references - there is no numerical order of references and also they are mentioned by their names throughout the text.

Comments on the Quality of English Language

The quality of the English language is generally fine except for minor errors.

Author Response

# Reviewer 2

Dear authors,

Despite the interesting topic provided in the title, unfortunately there are way too serious issues with the manuscript:

  1. In the introductions there is a lack of a problem presented so that the current research is justified. In order to prove your point you should provide enough robust evidence that the microbiota and associated changes of metabolites lead to any changes in early pregnancy, and that further studies in that topic are needed.

Response: Thanks for this comment, according to your suggestion, we have rephrased the contents in the part of “Introduction”.

  1. As a practicing clinician my biggest problem is that there is absolutely no discussion or evidence on how would this research translate to clinical practice. 

Response: Thank you for the reviewer's suggestions. We have added a discussion about future clinical research in the discussion section.

  1. The results section does not explain how the findings may be correlated to changes in early pregnancy and what if any interventions would be able to affect them.

Response: Thank you for this comment. In response to the reviewer’s suggestions, strategies for potential interventions during early pregnancy, based on the identified metabolites and gut microbes, have been added to the manuscript.

  1. You have not followed MDPI's guidelines on citing references - there is no numerical order of references and also they are mentioned by their names throughout the text.

Response: Thanks for this suggestion. We have changed the style of reference in order to meet the requirement in MDPI’s guidelines.

Reviewer 3 Report

Comments and Suggestions for Authors

This manuscript is a confused and disjointed attempt at exploring the relationship between gut microbiota, metabolism, and transcriptomics during early pregnancy. While the topic is potentially interesting, the execution is poor, and the manuscript is riddled with issues that undermine its scientific validity. The data presentation is sloppy, the methodology lacks critical details, and the discussion is superficial at best. Significant revisions are needed before this manuscript can be considered for publication.

The methodology section is a mess. Important details are missing, and there is a lack of transparency regarding the experimental procedures. How were the animals selected? What were the exclusion criteria? These are basic questions that the authors failed to address. The statistical analysis is inadequately described, leaving the reader to question the reliability of the results. Without proper controls and a clear explanation of the methods, the validity of the findings is highly questionable.

This manuscript is far from ready for publication. The authors need to address the numerous issues raised in this review, including clarifying their methodology, improving data presentation, and providing a more thoughtful discussion. Only after these major revisions should the manuscript be considered for further review.

Comments on the Quality of English Language

Numerous instances of subject-verb disagreement, incorrect tense usage, and improper article placement are throughout the manuscript. These basic errors indicate a lack of attention to detail and should be corrected.

Author Response

# Reviewer 3

This manuscript is a confused and disjointed attempt at exploring the relationship between gut microbiota, metabolism, and transcriptomics during early pregnancy. While the topic is potentially interesting, the execution is poor, and the manuscript is riddled with issues that undermine its scientific validity. The data presentation is sloppy, the methodology lacks critical details, and the discussion is superficial at best. Significant revisions are needed before this manuscript can be considered for publication.

Response: We really appreciate the reviewer’s comments. We have revised the manuscript thoroughly as advised by you. We believed that the manuscript has been improved. Revisions in the manuscript were highlighted in blue. Here, we provide a point-to-point response to your major and specific comments below.

The methodology section is a mess. Important details are missing, and there is a lack of transparency regarding the experimental procedures. How were the animals selected? What were the exclusion criteria? These are basic questions that the authors failed to address. The statistical analysis is inadequately described, leaving the reader to question the reliability of the results. Without proper controls and a clear explanation of the methods, the validity of the findings is highly questionable.

Response: Thanks for this instructive comment. According to your suggestion, we revised the part of “Materials and Methods”. Please see Lines 87-201.

This manuscript is far from ready for publication. The authors need to address the numerous issues raised in this review, including clarifying their methodology, improving data presentation, and providing a more thoughtful discussion. Only after these major revisions should the manuscript be considered for further review.

Response: Thanks for this comment, we revised this part of contents in the revised manuscript accordingly.

Numerous instances of subject-verb disagreement, incorrect tense usage, and improper article placement are throughout the manuscript. These basic errors indicate a lack of attention to detail and should be corrected.

Response: Thanks for these instructive comments regarding the improvement of the quality of this manuscript. According to the reviewer’s explanation, we have revised the relevant contents in every part in the manuscript.

Round 2

Reviewer 2 Report

Comments and Suggestions for Authors

Dear authors,

Thank you for answering and incorporating all my comments in the revised manuscript. I have no further issues.

Reviewer 3 Report

Comments and Suggestions for Authors

Ok good manuscript

Comments on the Quality of English Language

Minor errors